# ZnO-Au*_x_*Cu_1−*x*_ Alloy and ZnO-Au*_x_*Al_1−*x*_ Alloy Vertically Aligned Nanocomposites for Low-Loss Plasmonic Metamaterials

**DOI:** 10.3390/molecules27061785

**Published:** 2022-03-09

**Authors:** Robynne L. Paldi, Juanjuan Lu, Yash Pachaury, Zihao He, Nirali A. Bhatt, Xinghang Zhang, Anter El-Azab, Aleem Siddiqui, Haiyan Wang

**Affiliations:** 1School of Materials Engineering, Purdue University, West Lafayette, IN 47907, USA; rpaldi@purdue.edu (R.L.P.); lu790@purdue.edu (J.L.); ypachaur@purdue.edu (Y.P.); bhatt8@purdue.edu (N.A.B.); xzhang98@purdue.edu (X.Z.); aelazab@purdue.edu (A.E.-A.); 2School of Electrical and Computer Engineering, Purdue University, West Lafayette, IN 47907, USA; he468@purdue.edu; 3Sandia National Laboratory, Albuquerque, NM 87123, USA; asiddiq@sandia.gov

**Keywords:** ZnO, Au, vertically aligned nanocomposite, low loss, metamaterials, Cu, Al

## Abstract

Hyperbolic metamaterials are a class of materials exhibiting anisotropic dielectric function owing to the morphology of the nanostructures. In these structures, one direction behaves as a metal, and the orthogonal direction behaves as a dielectric material. Applications include subdiffraction imaging and hyperlenses. However, key limiting factors include energy losses of noble metals and challenging fabrication methods. In this work, self-assembled plasmonic metamaterials consisting of anisotropic nanoalloy pillars embedded into the ZnO matrix are developed using a seed-layer approach. Alloys of Au*_x_*Al_1−*x*_ or Au*_x_*Cu_1−*x*_ are explored due to their lower losses and higher stability. Optical and microstructural properties were explored. The ZnO-Au*_x_*Cu_1−*x*_ system demonstrated excellent epitaxial quality and optical properties compared with the ZnO-Au*_x_*Al_1−*x*_ system. Both nanocomposite systems demonstrate plasmonic resonance, hyperbolic dispersion, low losses, and epsilon-near-zero permittivity, making them promising candidates towards direct photonic integration.

## 1. Introduction

Hybrid plasmonic metamaterials are materials artificially constructed with more than one material, with plasmonic properties as well as other exotic properties, resulting from the constructed hybrid structures. Some of the exotic optical functionalities include hyperbolic dispersion, epsilon-near-zero (ENZ) permittivity and nanoscale light manipulation. These properties are created in anisotropic nanostructures in which in one direction it behaves as a dielectric material and in the perpendicular direction it behaves as a metal [1]. The periodic nature of the material enables a photonic band gap, whereby certain wavelengths of light are inhibited from propagation. Such a nanostructure can be achieved by directly embedding metallic nanowire into a dielectric medium [2,3,4]. Direct integration of hyperbolic metamaterials to optical devices has been predicted to allow for subdiffraction imaging [5] and waveguides [6], with new insight into applications in high Tc superconductivity [7] and for studying fundamental physical phenomena [8,9].

Major issues inhibiting the device applications of hybrid plasmonic metamaterials include materials processing and high absorption losses due to the metallic constituent. Methods for developing nanowire hyperbolic metamaterials include anodic aluminum oxide (AAO) template [10,11] and e-beam lithography [12]. Though promising, these methods suffer from materials selection issues and tedious multistep fabrication. As an alternative approach, a direct growth method could allow for the direct on-chip integration of hyperbolic metamaterials for device applications. Moreover, in the case of nanowire hyperbolic metamaterials, controlling the morphology of the nanostructure such as anisotropy and in-plane ordering are proven to be efficacious for controlling optical properties [1]. Another critical challenge for device integration is the occurrence of joule absorption loss caused by the use of plasmonic noble metals such as Ag or Au. Other plasmonic metals such as Al [13] and Cu [14] have been proposed as alternatives for noble metals, but they are difficult to incorporate due to low thermal stability. 

Recently, great interest has arisen in oxide–metal vertically aligned nanocomposite (VAN) thin films for use as hyperbolic metamaterials [15]. VAN thin films are made up of two immiscible materials, co-deposited during a one-step self-assembly pulsed-laser deposition (PLD) technique [16,17,18,19]. The resultant morphology enables the formation of a matrix phase of one material embedded within the pillars of the second phase. VAN thin films allow for the exotic and tunable combination of multifunctionalities including electrical [20], optical [15], magnetic [21], etc. The benefit of the VAN as a growth method for hyperbolic metamaterials include the breadth of dielectric matrix and metal pillar selection and the direct, one-step fabrication. Previously reported oxide–metal VAN for metamaterial application include ZnO-Au [22], ZnO-Cu [23], BTO-Au [15], ZnO-Au_*x*_Ag_1−*x*_ [24], and more. Prior studies with ZnO-metal systems have used different concentrations of metals which show the ease at which ZnO can be doped with metals. Nanocomposite targets composed of concentrations ranging from 1:1 Au/ZnO [22] to 9:1 ZnO/Au, 9:1 ZnO/Cu, and 9:1 ZnO/Al [25] have all been utilized. ZnO was chosen considering its chemical inertness and robustness during aging. In addition, prior in situ and ex situ annealing studies have suggested the robust thermal stability in these ZnO-metal nanocomposites [26] and other oxide–metal nanocomposite thin films with superior thermal stability in structural and optical properties up to 900 °C annealing [27]. Moreover, the in-plane ordering of the pillars can be tuned and has been shown to be able to manipulate the hyperbolic properties in ZnO-Au VAN through the mesoscale quasi-hexagonal in-plane arrangement by systematic changes in the deposition conditions [22]. Recently, the oxide–nanoalloy VAN of ZnO-Au_*x*_Ag_1−*x*_ and the use of plasmonic nanoalloys in replacement of noble metal Au or Ag has been demonstrated to lower absorption losses in hyperbolic metamaterials [24]. Utilization of the VAN platform offers the benefit of providing a highly tunable system through morphology, access to multifunctionalities through materials selection, a one-step self-assembly, and a pathway for lowering losses by using nanoalloys.

In this work, VAN of low-loss oxide–nanoalloy films are demonstrated. ZnO-Au*_x_*Cu_1−*x*_ and ZnO-Au*_x_*Al_1−*x*_ are developed as low-loss plasmonic metamaterials through a seed layer approach that helps to facilitate pillar growth and alloying, as illustrated in Figure 1. Cu and Al metals were selected for this study due to their low absorption losses and plasmonic response [14,24]. ZnO was chosen for the matrix due to its piezoelectric and dielectric response and general availability [28,29]. The use of a nanoalloy is expected to lower the losses of ZnO-Au VAN while providing a platform to stabilize the growth of Al and Cu into the matrix. The seed layer of ZnO-Au provides the basis for directed and ordered growth. Detailed microstructural characterization was performed through transmission electron microscopy (TEM), X-ray diffraction (XRD), and scanning transmission electron microscopy (STEM). Optical characterization was performed using UV-Vis spectroscopy and spectroscopic ellipsometry. The correlation function was used to investigate the in-plane ordering and we compared it to that of ZnO-Au thin films. 

## 2. Results

ZnO-Au*_x_*Cu_1−*x*_ and ZnO-Au*_x_*Al_1−*x*_ nanocomposite thin films were grown through a two-step pulsed laser deposition. A seed-layer of ZnO-Au VAN [22] was utilized to help facilitate successful growth, where a composite seed-layer approach has been shown to promote high-quality single domain-like LiNbO_3_ thin-film growth [20]. The two-step seeded growth is necessary to promote ordered VAN growth of the new alloy system. Initial attempts were made to directly grow ZnO-Au*_x_*Cu_1−*x*_, and ZnO-Au*_x_*Al_1−*x*,_ which can be seen in Appendix A, respectively. In these depositions, alloyed Au*_x_*Cu_1−*x*_ nanoparticles were formed with low presence of ZnO as seen from the elemental energy dispersive (EDS) mapping. It is possible that in the case when a direct growth method is used, the Au*_x_*Cu_1−*x*_ nanoparticles nucleate and grow faster than that of ZnO. Due to this, Au*_x_*Cu_1−*x*_ nanoparticles have a lower activation energy for nucleation. A similar nanoparticle morphology resulted in the direct growth of ZnO-Au*_x_*Al_1−*x*_ as seen in Appendix A. A seed layer of ZnO-Au VAN was demonstrated to be necessary to promote the growth of alloyed nanopillar thin films. 

Microstructural characterization was performed on the ZnO-Au*_x_*Cu_1−*x*_ thin film with the cross-section depicted in Figure 2. In Figure 2a, a representative cross-section scanning transmission electron microscopy (STEM) under the high-angle angular dark field (HAADF) mode is shown. HAADF-STEM provides a strong contrast between the pillar and matrix due to the atomic number scaling (~Z^1.7^). A corresponding selective area electron diffraction (SAED) pattern is depicted in Figure 2b. The SAED pattern was taken along the <11–20> sapphire zone axis and shows the in-plane and out-of-plane epitaxial relationship between the film and substrate. For out-of-plane, the relationship is Al_2_O_3_ (0006) // Au*_x_*Cu_1−*x*_ (111) // ZnO (0002) and for in-plane the epitaxial relationship is Al_2_O_3_ $\left(03\bar{3}0\right)$ // Au*_x_*Cu_1−*x*_ $\left(20\bar{2}\right)$ // ZnO $\left(11\bar{2}0\right)$. Energy-dispersive spectroscopy was utilized to determine if the pillar constituents formed an alloy solution. The mapping for the nanopillar alloy constituents of Au and Cu are depicted in Figure 2c,d, respectively. The corresponding HAADF image is shown in Figure 2a. Au and Cu mix very well and form an alloyed pillar as the EDS mapping is concentrated in the pillar location. Interesting to note in the EDS mapping of the nanopillars is that the Cu diffuses down into the seed layer and an alloyed pillar phase is formed in the matrix throughout the pillar thickness. The mapping for the ZnO matrix is depicted in Figure 2e, with the matrix and pillar forming two distinct phases with no interdiffusion. The combined map for nanoalloy pillars and oxide matrix is shown in Figure 2f. X-ray diffraction scans of θ–2θ were performed on the ZnO-Au*_x_*Cu_1−*x*_ and ZnO-Au*_x_*Al_1−*x*_ VAN and are depicted in Appendix A. The Cu nanoalloy XRD scan displays out-of-plane epitaxial relationship of Al_2_O_3_ (0006) // Au*_x_*Cu_1−*x*_ (111) // ZnO (0002), confirming the epitaxial relationship in the SAED pattern. Moreover, the Al nanoalloy displays a similar epitaxial relationship of Al_2_O_3_ (0006) // Au*_x_*Al_1−*x*_ (111) // ZnO (0002), indicating that the ZnO-Au forms a platform for epitaxial growth of oxide–nanoalloy VAN. 

The plan-view analysis of the ZnO-Au*_x_*Cu_1−*x*_ hybrid film was also conducted and is shown in Figure 3. The HAADF image is shown in Figure 3a, depicting the Au*_x_*Cu_1−*x*_ nanopillars arranged with mesoscale in-plane hexagonal ordering, which is a result of the ZnO-Au seed layer [22]. In the HRTEM, the ZnO matrix and nanoalloy pillars grow with hexagonal in-plane epitaxy. Moreover, the alloy pillar grows with a face-centered cubic structure, as seen from the atomic arrangement and hexagonal shape of the pillars in Figure 3b. The FCC Au rotates in the matrix to expose the (111) out-of-plane orientation and grow epitaxially with the wurtzite ZnO matrix. The atomic arrangement seen in the plan-view HRTEM micrograph is consistent with the epitaxial relationship in the cross-section SAED pattern in Figure 2b. EDS elemental mapping of the plan-view sample was also performed to determine if there was any diffusion between the matrix and pillar. The mapping of the Au and Cu metal can be seen in Figure 3c,d, respectively. Elemental mapping of Zn is shown in Figure 3e and the combined map is shown in Figure 3f, all corresponding to the plan-view HAADF in Figure 3a. Results of the in-plane EDS mapping confirm no interdiffusion between the matrix and nanopillars. Mapping of the Cu and Au overlap at the location of the pillar, indicating the formation of alloyed pillars. The two-step growth with the ZnO-Au seed layer was utilized for developing ZnO-Au*_x_*Al_1−*x*_ VAN with results of cross-section STEM and EDS mapping depicted in Appendix A. Plan-view STEM and EDS mapping are shown in Appendix A. The accompanying discussion for the growth of ZnO-Au*_x_*Al_1−*x*_ nanocomposite can be found in the Appendix A. 

One of the purposes of this study is to seek low-loss plasmonics and hyperbolic metamaterials. Optical properties were explored through both UV-Vis spectroscopy and Spectroscopic Ellipsometry, and results of the UV-Vis spectroscopy measurement are shown in Figure 4 with measurements performed on both ZnO-Au*_x_*Cu_1−*x*_ and ZnO-Au*_x_*Al_1−*x*_. Depicted in Figure 4a are the UV-Vis measurements for both films of ZnO-Au*_x_*Cu_1−*x*_ and ZnO-Au*_x_*Al_1−*x*_, while the results of ellipsometry are shown in Figure 4b–e. In the UV-Vis data, plasmon resonance dips were seen at 562 nm for ZnO-Au*_x_*Cu_1−*x*_ and at 586 nm for ZnO-Au*_x_*Al_1−*x*_. For the ellipsometry measurement, data were measured in the range of 210–2500 nm for each film and the collected data psi and delta are plotted in Appendix A. Anisotropic complex dielectric function was derived from measured *ψ* and *Δ* using a uniaxial coupled with b-spline model, and the experimental setup of the experiments are inset to Figure 4c for the ZnO-Au*_x_*Cu_1−*x*_ film and Figure 4e for the ZnO-Au*_x_*Al_1−*x*_ film. Around the plasmonic resonance wavelength range, for real permittivity that represents the refraction index, there should be a corresponding intensity dip. Additionally, for the imaginary part of the dielectric constant, an intensity peak that implies an increase in the optical absorption loss should be found [7]. It should be noted that there is a slight difference between the plasmonic wavelength locations based on the ellipsometry plots, which could be related to the selection of an anisotropic thin-film model instead of a homogeneous one in the optical fitting process. A similar finding can be found in a previous report [30].

The real part of the permittivity for the ZnO-Au*_x_*Cu_1−*x*_ is depicted in Figure 4b, demonstrating hyperbolic dispersion in the near-infrared regime with an epsilon-near-zero (ENZ) point at around 1200 nm. The ZnO-Au*_x_*Cu_1−*x*_ film demonstrates type 1 hyperbolicity, indicating that ε_//_ > 0 and ε_⊥_ < 0; the representative isofrequency curve inset to Figure 4b is a hyperboloid of one sheet, which can support high- and low-k photonic density states [2]. The imaginary permittivity, which relates to the amount of light absorbed into the film, is presented in Figure 4c. Compared to other ZnO-metal structures (e.g., Ag/ZnO layered structure, Al/ZnO layered structure, and ZnO-Ag*_x_*Au_1−*x*_ hyperbolic metamaterial films) [24,31,32], the ZnO-Au*_x_*Cu_1−*x*_ film presents much lower values of imaginary permittivity, indicating that using Cu alloy is an efficacious approach for reducing optical loss in hyperbolic metamaterials. Specifically, around the plasmonic wavelength, the peaks in the imaginary permittivity ranges from 2 to 5, compared to 3 to 18 in the case of ZnO-Au*_x_*Ag_1−*x*_ VAN films. Additionally, the absorption is constant and stable for both ZnO-Au*_x_*Cu_1−*x*_ and ZnO-Au*_x_*Al_1−*x*_ across UV to NIR region, compared to the previous report of ZnO-Au that has an imaginary permittivity of over 45 in the higher wavelength region [24]. The real part of permittivity for the ZnO-Au*_x_*Al_1−*x*_ film is depicted in Figure 4d and imaginary permittivity is presented in Figure 4e. ZnO-Au*_x_*Al_1−*x*_ film demonstrated type 1 hyperbolicity in a much smaller range, with the ENZ permittivity point occurring much farther in the near-IR range, at around 2250 nm. Moreover, the imaginary permittivity values are slightly lower than those of the ZnO-Au*_x_*Cu_1−*x*_. Overall, the ZnO-Au*_x_*Cu_1−*x*_ presents much better properties. For both films, in the visible wavelength region, though the real parts of the dielectric functions are positive, they tend to be tuned negative, especially in the Au*_x_*Cu_1−*x*_ film. This tunability is highly promising based on the versatile oxide–metal VAN microstructures and their close relationship with the corresponding optical properties.

In-plane ordering has been shown to be important in controlling optical properties in optical metamaterials [1,5,33]. To directly investigate the in-plane ordering of VAN films, we determined the two-point correlation of the alloyed nanopillars. The ZnO-Au*_x_*Cu_1−*x*_ film was selected for this study due to the ordered structure and was compared with a ZnO-Au VAN film. The two-point correlation function describes the probability of finding two distinct local states at the endpoints of a vector in a random direction in the nanostructure, as given by Equation (1) [34,35]
(1)$$f_{r}^{ij}=\frac{1}{S}\overset{S-1}{\underset{x_{i}=0}{\sum }}m_{x_{i}}^{i}m_{x_{i}+r}^{j}$$

In this equation, *S* represents the total number of grid points, $m_{x_{i}}^{i}$ represents the microstructure function detailing the probability of finding local state *i* in the nearest neighbor of a point *x_i_*. $m_{x_{i}+r}^{j}$ relates to the probability of finding local state *j* at a distance *r* from *x_i_*. A local state is defined as the localized composition of the material being studied, i.e., the matrix and pillars. The results of the correlation function calculation for ZnO-Au and ZnO-Au*_x_*Cu_1−*x*_ are depicted in Figure 5. Figure 5a shows the reference STEM image of ZnO-Au, Figure 5b shows the binarized image of Figure 5a, and Figure 5c shows the correlation function. ZnO-Au has both long-range and short-range order, with short-range taking on a six-fold symmetry, as predicted in the previous literature [22]. In Figure 5d is shown the reference STEM image for the ZnO-Au*_x_*Cu_1−*x*_ film, in Figure 5e is shown the binarized image of Figure 5d, and the correlation function is shown in Figure 5f. The ZnO-Au*_x_*Cu_1−*x*_ film also possesses long-range and short-range order, though interestingly, the in-plane ordering has an eight-fold symmetry. The tuning of the in-plane ordering reveals that alloying can possibly serve as a tool for tuning spontaneous ordering in oxide–metal VAN, which is important for tailoring specific optical properties.

## 3. Discussion

Overall, the ZnO-Au*_x_*Cu_1−*x*_ film made through a two-step seeded growth involving a ZnO-Au seed layer presents a better morphology and optical properties, while both films demonstrated plasmon resonance in the visible regime. The seed-layer approach could be used to realize other oxide–nanoalloy films that suffer during direct growth from nucleation and the growth issues seen in the ZnO-Au*_x_*Cu_1−*x*_ and ZnO-Au*_x_*Al_1−*x*_, i.e., they tend to form islands during direct growth. Other examples could include magnetic material alloys such as Ni, Co, and Fe or other optical materials such as Pt, Ti, etc., or any multifunctional combination thereof. The films in this report demonstrate intriguing optical properties that can be used to realize potential applications towards the future integration of nanoalloy-based optics with a tailorable hyperbolic wavelength range. It has been reported that point defects could play an important role in the overall optical properties of ZnO [36]. Considering the similar growth conditions for the ZnO films in this study, the concentration of the point defects could be comparable in the samples. Thus, this study focused more on the tuning of the pillar geometry and distribution. The detailed point defect analysis in these ZnO-metal hybrid metamaterials could be an interesting topic for future studies. 

## 4. Materials and Methods

### 4.1. Thin-Film Growth

ZnO-Au*_x_*Cu_1−*x*_ and ZnO-Au*_x_*Al_1−*x*_ nanocomposite films were grown on c-cut Al_2_O_3_ (0001) using pulsed-laser deposition through a two-step seeded growth method. In the first step, a ZnO-Au seed layer was deposited and in the second step a ZnO-Au nanocomposite target with either Cu or Al metal strip glued to it was used. The ZnO-Au composite target was developed through solid-state sintering. Growth was performed with a KrF excimer laser (Lambda Physik Complex Pro 205, λ = 248 nm) and substrate temperature was kept constant at 500 °C. The laser energy was 420 mJ focused at an incident angle of 45°. The target–substrate distance was kept constant at 4.5 cm and measured before each deposition to ensure accuracy. Before deposition, the chamber was pumped down to around 10^−6^ mTorr before an oxygen pressure was inflowed. The deposition was performed in vacuum and the laser pulse frequency was set to 5 Hz for all deposition. After all depositions, the chamber was cooled to room temperature at a rate of 15 °C/min.

### 4.2. Microstructure Characterization

Film morphology was characterized through XRD, TEM, and STEM coupled with EDS mapping. XRD scans of θ–2θ were conducted using a Panalytical X’Pert X-ray diffractometer with Cu K_α_ radiation. Bright-field TEM, STEM, SAED patterns, and EDS mapping were performed in a FEI Talos F200X TEM. Samples for electron microscopy were prepared, for both cross-section and plan-view, via a standard grinding procedure which entails manual grinding, polishing, dimpling, and a final ion milling step to achieve electron transparency (PIPS 691 precision ion polishing system, 5 KeV for cross-section and 4–4.5 KeV for plan-view sample).

### 4.3. Optical Property Measurements

Ellipsometry experiments were carried out using an RC2 Spectroscopic ellipsometer (J.A. Woollam Company, Lincoln, NE, USA). Three angles, 30°, 45°, and 60°, were measured from a spectrum range of 210–2500 nm. Measured values of psi and delta data were obtained and then fit with a uniaxial model coupled with a B-spline model to discern the anisotropic dielectric function of ZnO-Ag*_x_*Au_1−*x*_ nanocomposite thin films. The uniaxial model provides both in-plane and out-of-plane dielectric functions and a B-spline model was used in each direction to obtain an agreeable mean square error (MSE) < 5.

## 5. Conclusions

A two-step seed layer approach has enabled the growth of low-loss plasmonic oxide–nanoalloy VAN systems, including ZnO-Au*_x_*Cu_1−*x*_ and ZnO-Au*_x_*Al_1−*x*_. The seed layer approach enables alloy pillar formation instead of the nanoparticle formation in the one-step growth without the seed layer. The ZnO-Au*_x_*Cu_1−*x*_ film was determined to have superior morphology to the ZnO-Au*_x_*Al_1−*x*_ due to the formation of Al-doped ZnO during the second step of the growth. Both films demonstrated plasmonic response in the visible regime and showed hyperbolic dispersion. Moreover, the films in this report have much lower absorption losses than those of the previously reported oxide–metal VAN. The in-plane ordering of the ZnO-Au*_x_*Cu_1−*x*_ film was calculated using a two-point correlation function and compared with ZnO-Au VAN. ZnO-Au*_x_*Cu_1−*x*_ was found to have an eight-fold in-plane symmetry, revealing that alloying Cu into ZnO-Au causes a shift in the in-plane ordering and provides a way to tune ordering of the VAN and potentially the optical properties. The work in this report opens the pathway towards future studies of oxide–nanoalloy VAN for optical applications requiring a tailorable hyperbolic wavelength range.

## Figures and Tables

**Figure 1 molecules-27-01785-f001:**
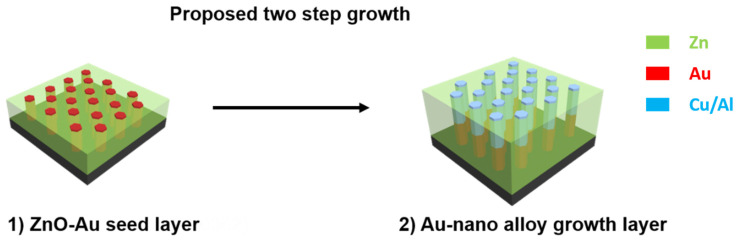
Two-step seeded growth approach was utilized to allow for the formation of low-loss oxide–alloy nanocomposites.

**Figure 2 molecules-27-01785-f002:**
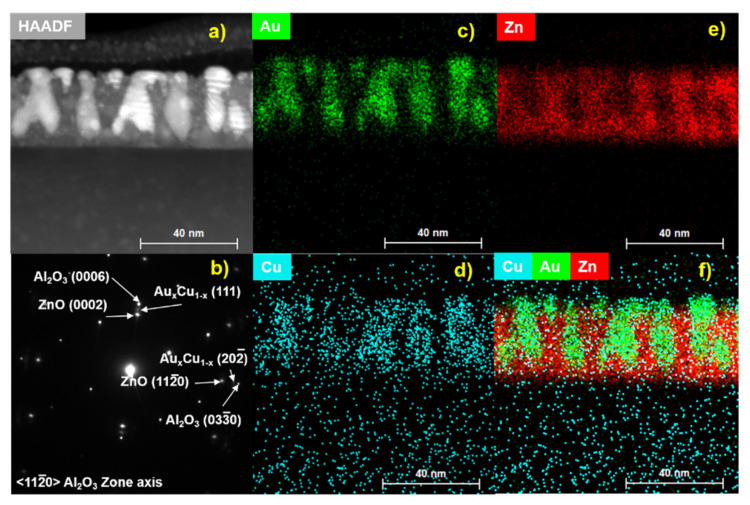
Cross-section STEM image of ZnO-Au_*x*_Cu_1−*x*_. (**a**) HAADF cross-section image, (**b**) cross-section SAED pattern, (**c**) EDS Au map, (**d**) EDS Cu-map, (**e**) EDS Zn-map, (**f**) combined EDS-map.

**Figure 3 molecules-27-01785-f003:**
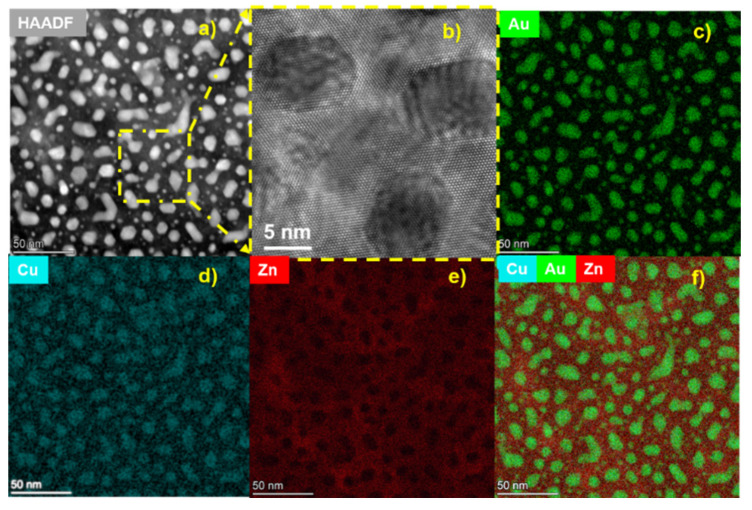
Plan view of ZnO-Au*_x_*Cu_1−*x*_. (**a**) STEM plan view of ZnO-Au*_x_*Cu_1−*x*_, (**b**) high-resolution TEM image of yellow box inset in (**a**). Elemental EDS mapping of (**c**) Au, (**d**) Cu, (**e**) Zn, and (**f**) combined map.

**Figure 4 molecules-27-01785-f004:**
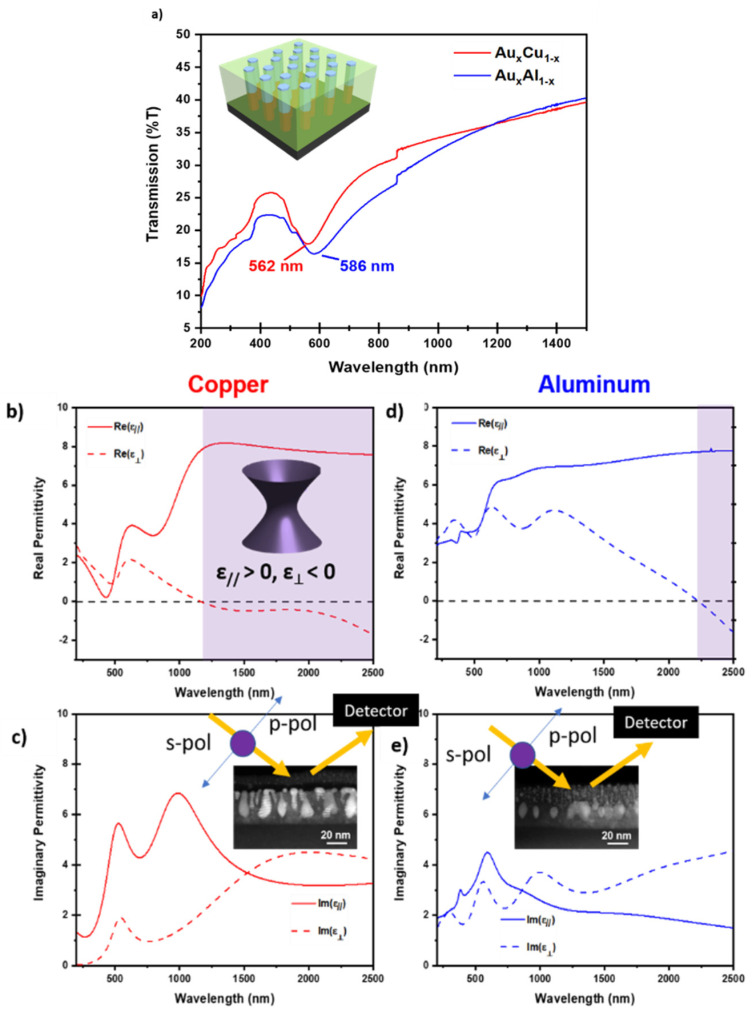
Optical measurements. (**a**) Normal incident depolarized transmittance (T%) of ZnO-Au*_x_*Cu_1−*x*_ and ZnO-Al*_x_*Au_1−*x*_. (**b**) Real part of permittivity for ZnO-Au*_x_*Cu_1−*x*_. Highlighted in purple is the location of the hyperbolic regime and the inset diagram represents the isofrequency curve of a. (**c**) Imaginary permittivity for ZnO-Au*_x_*Cu_1−*x*_. Inset is the experimental setup for the ellipsometry measurement for ZnO-Au*_x_*Cu_1−*x*_. (**d**) Real permittivity for ZnO-Au*_x_*Al_1−*x*_. Highlighted in purple is the representative hyperbolic regime. (**e**) Imaginary permittivity for ZnO-Au*_x_*Cu_1−*x*_. Inset is the experimental setup for the ellipsometry measurement with respect to the ZnO-Au*_x_*Al_1−*x*_ thin film.

**Figure 5 molecules-27-01785-f005:**
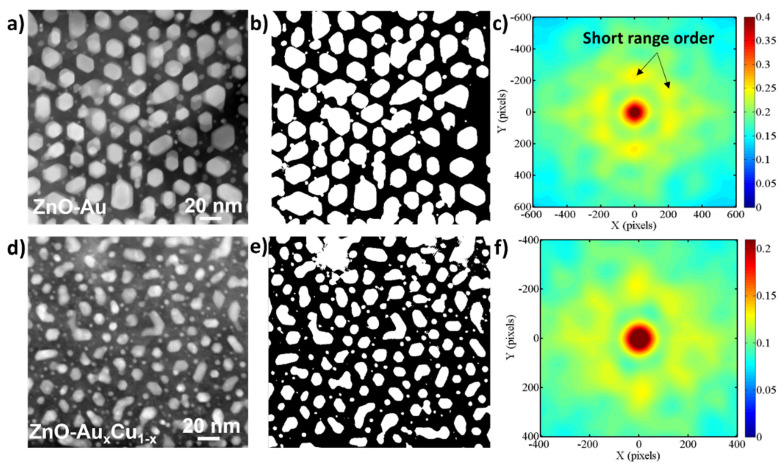
Correlation function. (**a**) STEM plan view of ZnO-Au. (**b**) Binarized image of plan-view ZnO-Au in (**a**). (**c**) Correlation function of plan-view in (**a**). (**d**) STEM plan view of ZnO-Au*_x_*Cu_1−*x*_. (**e**) Binarized image of plan view ZnO-Au*_x_*Cu_1−*x*_ in (**d**). (**f**) Correlation function of ZnO-Au*_x_*Cu_1−*x*_.

## Data Availability

Research data is available upon request to the corresponding author at hwang00@purdue.edu.

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
