# Peer review of "ZnO-AuxCu1−x Alloy and ZnO-AuxAl1−x Alloy Vertically Aligned Nanocomposites for Low-Loss Plasmonic Metamaterials"

_molecules, 2022, doi:10.3390/molecules27061785_

Round 1
Reviewer 1 Report
The authors presented the fabrication and test of the ZnO-AuxCu1-x materials system, the idea is interesting, and the writing is acceptable. Thus, I would like to suggest the acceptance of this manuscript after mino revision of some small grammar errors
Author Response
We greatly appreciate your careful consideration and valuable comments to our submission. All the comments serve very well in improving the quality of this paper. We have made all the revisions accordingly and highlighted below for reference. The major revisions include:
- We have revised our discussion section on the optical properties, including the interpretation of plasmonic dip and low-loss plasmonic properties;
- We have included model parameters for the VASE data;
- We have improved the figures labeling and captions for better readers experience;
- We have added discussion on point defects and aging of the samples;
- We have also conducted typo correction and revisions to the entire manuscript;
The point-by-point replies are summarized below for your reference. The revised manuscript is also highlighted for all the revisions.
The authors presented the fabrication and test of the ZnO-AuxCu1-x materials system, the idea is interesting, and the writing is acceptable. Thus, I would like to suggest the acceptance of this manuscript after minor revision of some small grammar errors.
Authors’ Reply:
We deeply appreciate the comments of this reviewer. After reviewing this manuscript again, we have made grammatical fixes where it seemed necessary. Some examples of the revision are highlighted below:
Revised manuscript pg. 3:
In the HRTEM, the ZnO matrix and nanoalloy pillars growth with hexagonal in-plane epitaxy. Moreover, the alloy pillar grows with a face-centered cubic structure as seen from the atomic arrangement and hexagonal shape of the pillars in Figure 3b.
Revised manuscript pg. 4:
One of the purposes of this study is seeking to seek low loss plasmonics and hyperbolic metamaterials.
Revised manuscript pg. 5:
In Figure 5a is shown shows the reference STEM image of ZnO-Au, Figure 5b shows the binarized image of Figure 5a,
Revised manuscript pg. 7:
Figure 3 Plan-view of ZnO-AuxCu1-x. a) STEM plan-view of ZnO-AuxCu1-x, b) high-resolution TEM image of yellow box inset in a). Elemental EDS mapping of c) Au, d) Cu, e) ZnO, and f) combined map.
Figure 4 c) Imaginary permittivity for ZnO-AuxCu1-x. Inset is the experimental setup for the ellipsometry measurement for ZnO-AuxCu1-x. d.) Real permittivity for ZnO-AuxAl1-x. Highlighted in purple is the representative hyperbolic regime. e.) Imaginary permittivity for ZnO-AuxCu1-x. Inset is the experimental setup for the ellipsometry measurement with respect to the ZnO-AuxAl1-x thin film.
Reviewer 2 Report
The authors present a study of oxide-nanoalloy VAN systems and report the optical properties.
General comments on editing: The authors need to improve on clarity and proof read the manuscript. English editing is highly encouraged for better legibility, however certain obvious editing errors must be corrected (e.g. missing references in line 90). Wording using identifiers such as “unique” should be avoided.
Figures are sparsely captioned, more detail within the captions would improve the manuscript. E.g. Figure 4a) does not indicate any measurement parameters, only in the experimental section it is mentioned how the data was recorded. Figure 1 does not label the color scheme used (I realize it is the same as in the following figure but a proper legend will improve the readers experience). Figure 4 especially has hard to read components (axis, scale bars, etc...) that should be improved throughput all figures.
The authors make claims of “low-loss” plasmonic properties. Such a declaration is only meaningful when presented in the context of other materials and by quantifying the loss. A succinct description (e.g. in a table) of the optical properties of the presented material and the materials it is compared to can achieve this.
The VASE data needs significantly more elaboration. The authors present a series of VASE derived dielectric functions, which are the main result from this study, however, they do not present any of the model parameters, other than “uniaxial model coupled with B-spline model”. The actual model parameters need to be included in the supplement at least.
My main point of criticism however is regarding the interpretation of the optical data. The authors refer to a “plasmonic dip” in the VIS transmission data for their samples (fig 4a). However, in the same figure, the in - and out of plane - dielectric function for each material remains >0 at the energies of the “plasmonic dip”. How do the authors explain these discrepancies?
At lower energies, both materials do exhibit the discussed hyperbolic behavior with out of plane dielectric functions exhibiting ENZ behavior. At visible wavelengths however, according to the authors own model, the material behaves as a lossy dielectric.
In combination with the above editing, I recommend major revisions of the manuscript.
Author Response
We greatly appreciate your careful consideration and valuable comments to our submission. All the comments serve very well in improving the quality of this paper. We have made all the revisions accordingly and highlighted below for reference. The major revisions include:
- We have revised our discussion section on the optical properties, including the interpretation of plasmonic dip and low-loss plasmonic properties;
- We have included model parameters for the VASE data;
- We have improved the figures labeling and captions for better readers experience;
- We have added discussion on point defects and aging of the samples;
- We have also conducted typo correction and revisions to the entire manuscript;
The point-by-point replies are summarized below for your reference. The revised manuscript is also highlighted for all the revisions.

Reviewer 3 Report
Referee report on manuscript “ZnO-AuxCu1-x alloy and ZnO-AuxAl1-x alloy vertically-aligned nanocomposites for low-loss plasmonic metamaterials” by Robynne L Paldi et al.
This is a very good and important article that is suitable for publication after taking into account few small suggestion, which will certainly improve this manuscript.
- Line 26. The phrase “Hybrid plasmonic metamaterials” needs more disclosure of meaning in order to be immediately understandable to a wide range of readers
- Line 46 -59. Here it is imperative to say about the concentration of metals. The fact is that zinc oxide and many other oxides are easily doped with metals and readers should clearly see the difference here. See, some example for Cu, Ag and Au:
Sugak, D., Yakhnevych, U., Syvorotka, I. I., et al (2019). Optical investigation of the OH− groups in the LiNbO3 doped by copper. Integrated Ferroelectrics, 196(1), 32-38.
Feldbach, E., Kirm, M., Lushchik, A., et al (2000). Excitonic and electron-hole processes in NaCl and NaCl: Ag crystals under conditions of multiplication of electronic excitations. Journal of Physics: Condensed Matter, 12(9), 1991.
Tomaev, V. V., Polishchuk, V. A., & Vartanyan, T. A. (2019). Optical density of nanocomposite ZnO films doped with Au, Al, Cu. In AIP Conference Proceedings (Vol. 2064, No. 1, p. 040006).
- Whether it is possible to conclude from the obtained data of optical measurements whether the samples are free from point defects or point defects are seen in the spectra . See, what more details in
Uklein, A. V., Multian, V. V., Kuz'micheva, at all (2018). Nonlinear optical response of bulk ZnO crystals with different content of intrinsic defects. Optical Materials, 84, 738-747.
- Could you comment on whether there is aging of the samples during time? This is absolutely important for practical applications.
Author Response
We greatly appreciate your careful consideration and valuable comments to our submission. All the comments serve very well in improving the quality of this paper. We have made all the revisions accordingly and highlighted below for reference. The major revisions include:
- We have revised our discussion section on the optical properties, including the interpretation of plasmonic dip and low-loss plasmonic properties;
- We have included model parameters for the VASE data;
- We have improved the figures labeling and captions for better readers experience;
- We have added discussion on point defects and aging of the samples;
- We have also conducted typo correction and revisions to the entire manuscript;
The point-by-point replies are summarized below for your reference. The revised manuscript is also highlighted for all the revisions.
This is a very good and important article that is suitable for publication after taking into account few small suggestions, which will certainly improve this manuscript.
- Line 26. The phrase “Hybrid plasmonic metamaterials” needs more disclosure of meaning in order to be immediately understandable to a wide range of readers.
Authors’ Reply:
We greatly value the reviewer’s comment. As a result, we have amended and rewritten this specific part of the manuscript so that the phrase “Hybrid plasmonic metamaterials” can be better understood by a wide range of readers. The specific revision is written below:
Revised manuscript Pg. 1:
Hybrid plasmonic metamaterials are materials artificially constructed with more than one material and show plasmonic properties as well as other exotic properties, resulting from the constructed hybrid structures. Some of the exotic optical functionalities include hyperbolic dispersion, epsilon near zero (ENZ) permittivity and nanoscale light manipulation.
- Line 46 -59. Here it is imperative to say about the concentration of metals. The fact is that zinc oxide and many other oxides are easily doped with metals and readers should clearly see the difference here. See, some example for Cu, Ag and Au:
Sugak, D., Yakhnevych, U., Syvorotka, I. I., et al (2019). Optical investigation of the OH− groups in the LiNbO3 doped by copper. Integrated Ferroelectrics, 196(1), 32-38.
Feldbach, E., Kirm, M., Lushchik, A., et al (2000). Excitonic and electron-hole processes in NaCl and NaCl: Ag crystals under conditions of multiplication of electronic excitations. Journal of Physics: Condensed Matter, 12(9), 1991.
Tomaev, V. V., Polishchuk, V. A., & Vartanyan, T. A. (2019). Optical density of nanocomposite ZnO films doped with Au, Al, Cu. In AIP Conference Proceedings (Vol. 2064, No. 1, p. 040006).
Authors’ Reply:
We greatly appreciate the reviewer’s careful comment. Therefore, we have conducted typo corrections and revisions to the entire manuscript. We have also included some of the related ZnO-metal references suggested by the reviewer. Some examples of the revision are highlighted below:
Revised manuscript pg. 3:
In the HRTEM, the ZnO matrix and nanoalloy pillars growth with hexagonal in-plane epitaxy. Moreover, the alloy pillar grows with a face-centered cubic structure as seen from the atomic arrangement and hexagonal shape of the pillars in Figure 3b.
Revised manuscript pg. 4:
One of the purposes of this study is seeking to seek low loss plasmonics and hyperbolic metamaterials.
Revised manuscript pg. 5:
In Figure 5a is shown shows the reference STEM image of ZnO-Au, Figure 5b shows the binarized image of Figure 5a,
Revised manuscript pg. 7:
Figure 3 Plan-view of ZnO-AuxCu1-x. a) STEM plan-view of ZnO-AuxCu1-x, b) high-resolution TEM image of yellow box inset in a). Elemental EDS mapping of c) Au, d) Cu, e) ZnO, and f) combined map.
Figure 4 c) Imaginary permittivity for ZnO-AuxCu1-x. Inset is the experimental setup for the ellipsometry measurement for ZnO-AuxCu1-x. d.) Real permittivity for ZnO-AuxAl1-x. Highlighted in purple is the representative hyperbolic regime. e.) Imaginary permittivity for ZnO-AuxCu1-x. Inset is the experimental setup for the ellipsometry measurement with respect to the ZnO-AuxAl1-x thin film.
Revised Manuscript pg. 2:
Previously reported oxide-metal VAN for metamaterials application include ZnO-Au [22], ZnO-Cu [23], BTO-Au [15], ZnO-AuxAg1-x [24], and more. Prior studies with ZnO-metal systems have used different concentrations of metals which show the ease at which ZnO can be doped with metals. Nanocomposite targets composed of concentrations ranging from 1:1 Au/ZnO [22] to 9:1 Zn/Au, 9:1 Zn/Cu, and 9:1 Zn/Al [30] have all been utilized.
Revised Manuscript pg. 11:
[30] V. V. Tomaev, V. A. Polishchuk, and T. A.Vartanyan, “Optical density of nanocomposite ZnO films doped with Au, Al, Cu.” AIP Conf Proc (2019)
- Whether it is possible to conclude from the obtained data of optical measurements whether the samples are free from point defects or point defects are seen in the spectra. See, what more details in
Uklein, A. V., Multian, V. V., Kuz’micheva, et al (2018). Nonlinear optical response of bulk ZnO crystals with different content of intrinsic defects. Optical Materials, 84, 738-747.
Authors’ Reply:
We greatly appreciate the reviewer’s comment. We agree with the reviewer that point defects could be important for the overall optical properties in oxides. Since all the ZnO thin films were deposited under the same growth conditions, e.g., temperature, oxygen partial pressure and laser energy, the defect level in ZnO could be comparable. The major differences are the metallic pillars incorporated. We have now included more discussions to discuss the importance of the point defect analysis and suggest it as possible future topic for exploration. Some of the related revisions are below/
Revised manuscript Pg. 8, end of ‘Discussion’ section:
Overall, the ZnO-AuxCu1-x film made through a unique two-step growth involving a ZnO-Au seed layer presents a better morphology and optical properties, while both films demonstrated plasmon resonance in the visible regime. The seed-layer approach could be used to realize other oxide-nanoalloy films that suffer during direct growth from nucleation and the growth issues seen in the ZnO-AuxCu1-x and ZnO-AuxAl1-x, i.e., they tend to form island during the direct growth. Other example could include magnetic materials alloys such as Ni, Co, and Fe or other optical materials like Pt, Ti, etc. or any multifunctional combination thereof. The films in this report demonstrate unique optical properties that can be used to realize potential applications towards the future integration of nanoalloy based optics with tailorable hyperbolic wavelength range. It has been reported that point defects could play an important role in the overall optical properties of ZnO [36]. Considering the similar growth conditions for the ZnO films in this study, the concentration of the point defects could be comparable in the samples. Thus, this study focused more on the tuning of the pillar geometry and distribution. The detailed point defect analysis in these ZnO-metal hybrid metamaterials could be an interesting topic for future studies.
Revised manuscript Pg. 11:
[36] A.V. Uklein, V.V. Multian, G.M. Kuz'micheva, R.P. Linnik, V.V. Lisnyak, A.I. Popov, and V. Ya. Gayvoronsky, “Nonlinear optical response of bulk ZnO crystals with different content of intrinsic defects.” Opt. Mater., 84, 738-747 (2018).
- Could you comment on whether there is aging of the samples during time? This is absolutely important for practical applications.
Authors’ Reply:
We appreciate the great comment from the reviewer. Based on our previous ex situ and in situ high temperature annealing analysis of the oxide-metal nanocomposite thin films, the thermal stability of the nanocomposite is high and the structure remains intact up to the high temperature as high as 900 oC for hours. There was no obvious change in terms of film morphology and the optical properties. We have included some of the related discussions based on the previous thermal annealing studies.
Revised manuscript pg. 2:
Previously reported oxide-metal VAN for metamaterials application include ZnO-Au [22], ZnO-Cu [23], BTO-Au [15], ZnO-AuxAg1-x [24], and more. ZnO was chosen considering its chemical inertness and robustness during aging. In addition, prior in situ and ex situ annealing studies have suggested the robust thermal stability in these ZnO-metal nanocomposites [31] and other oxide-metal nanocomposite thin films with superior thermal stability in structural and optical properties up to 900 oC annealing [32]. Moreover, the in-plane ordering of the pillars can be tuned and has been shown to be able to manipulate the hyperbolic properties in ZnO-Au VAN through the mesoscale quasi-hexagonal in-plane arrangement by systematic changes in the deposition conditions [22].
Revised manuscript pg. 11:
[31] S Misra, D Zhang, P Lu, H Wang, “Thermal stability of self-assembled ordered three-phase Au–BaTiO3–ZnO nanocomposite thin films via in situ heating in TEM.” Nanoscale, 12, 46 (2020).
[32] D. Zhang, Z. Qi, J. Jian, J. Huang, X. L. Phuah, X. Zhang, and H. Wang, "Thermally Stable Au-BaTiO3 Nanoscale Hybrid Metamaterial for High-Temperature Plasmonic Applications." ACS Appl. Nano Mater., 3, 1431-1437 (2020).
Round 2
Reviewer 2 Report
I thank the authors for addressing my concerns and for reworking the respective sections.
Author Response
We very much appreciate the reviewer's time and effort in reviewing our paper once again and also thank for his/her positive comments on our revision.
Reviewer 3 Report
the authors have successfully improved the manuscript by responding constructively to all the comments of the reviewer so that the article can be recommended for publication
Author Response

(The authors gave the same response as above.)
